**Subject Category:**
Biology (whole organism)

behaviour/ecology

South Africa, Benguela, pelagic, benthic, environmental change, aerobic dive limit

**Author for correspondence:**
S. P. Kirkman
e-mail: spkirkman@gmail.com

# Dive behaviour and foraging effort of female Cape fur seals *Arctocephalus pusillus pusillus*

S. P. Kirkman[1,2], D. P. Costa[3], A.-L. Harrison[3,4], P. G. H. Kotze[1], W. H. Oosthuizen[1], M. Weise[5], J. A. Botha[2] and J. P. Y. Arnould[6]

[1]Oceans and Coastal Research, Department of the Environment, Forestry and Fisheries, Private Bag X4390, Cape Town 8000, South Africa
[2]Marine Apex Predator Research Unit (MAPRU), Institute for Coastal and Marine Research, Nelson Mandela University, Port Elizabeth 6031, South Africa
[3]Department of Ecology and Evolutionary Biology, University of California, Santa Cruz, CA 95060, USA
[4]Migratory Bird Center, Smithsonian Conservation Biology Institute, National Zoological Park, Washington, DC 20008, USA
[5]Office of Naval Research—Code 32, 875 North Randolph Street, Arlington, VA 22203-1995, USA
[6]School of Life and Environmental Sciences, Faculty of Science Engineering and Built Environment, Deakin University, Melbourne, Australia

SPK, 0000-0001-5428-7375; A-LH, 0000-0002-6213-1765; JPYA, 0000-0003-1124-9330

While marine top predators can play a critical role in ecosystem structure and dynamics through their effects on prey populations, how the predators function in this role is often not well understood. In the Benguela region of southern Africa, the Cape fur seal (*Arctocephalus pusillus pusillus*) population constitutes the largest marine top predator biomass, but little is known of its foraging ecology other than its diet and some preliminary dive records. Dive information was obtained from 32 adult females instrumented with dive recorders at the Kleinsee colony (29°34.17′ S, 16°59.80′ E) in South Africa during 2006–2008. Most dives were in the depth range of epipelagic prey species (less than 50 m deep) and at night, reflecting the reliance of Cape fur seals on small, vertically migrating, schooling prey. However, most females also performed benthic dives, and benthic diving was prevalent in some individuals. Benthic diving was significantly associated with the frequency with which females exceeded their aerobic dive limit. The greater putative costs of benthic diving highlight the potential detrimental effects to Cape fur seals of well-documented changes in the availability of epipelagic prey species in the Benguela.

# 1. Introduction

It is well known that top predators can play a critical role in marine ecosystem structure and dynamics through their effects on prey populations and ecosystem function [1,2]. Knowledge of their foraging ecology, including trophic interactions, feeding behaviour and the factors influencing them, is key to understanding ecosystem functioning and the effects that removal, depletion, growth or range shifts of such populations may have on ecosystems. In this regard, advances in wildlife tracking tools and animal-borne behaviour data loggers provide critical insights into the foraging behaviour of marine top predators [3–5]. While over the last few decades, there have been a plethora of studies employing such technologies on a wide variety of marine predators [4,6,7], detailed data are still lacking for numerous ecologically important species.

Whereas many marine top predator species including several shark and seabird species and some marine mammal species have been declining at rapid rates worldwide [8,9], other species have been in recovery following population depletion due to past overharvesting [10–13]. The Cape fur seal (*Arctocephalus pusillus pusillus*), the only pinniped endemic to the African continent, is a marine top predator that has shown considerable population recovery following previous over-exploitation [14]. The current population size for the Cape fur seal has been estimated at 1.5–2 million [14,15], and its breeding range extends from the southeast coast of South Africa to southern Angola [13]. The population constitutes a major component of the top predator biomass in the highly productive Benguela Current Upwelling System (the Benguela), which coincides with the majority of the population's breeding range [16].

The annual biomass consumption by the Cape fur seal population is estimated at *ca* 2 million tons [17]. It has a generalist diet that includes teleost fish and elasmobranchs, cephalopods, crustaceans and seabirds. Two-thirds of the population's consumption comprises commercially targeted species, including hakes (*Merluccius* spp.), horse mackerel (*Trachurus trachurus capensis*), sardine (*Sardinops sagax*), anchovy (*Engraulis encrasicolus*), chokka squid (*Loligo vulgaris reynaudii*) and West Coast rock lobster (*Jasus lalandii*) [18–20]. Consequently, there is ever-present concern regarding competition for resources between the seal population, which removes *ca* 2 million tons of marine organisms per year in South Africa and Namibia [17], and commercial fisheries [21,22]. This has been a primary motivation for numerous diet studies of the Cape fur seal, mostly with comparison to fisheries [17,18,22–24].

While the diet of the Cape fur seal has been well elucidated, and despite the obvious importance of its population in the Benguela, little is known of its foraging behaviour [25]. Such information is crucial for understanding how this top predator responds to environmental variability [26,27]. Indeed, information derived from dive behaviour data loggers such as the shapes, depths, durations and frequency of dives in relation to physiological limits and environmental variables are widely used to infer the responses of marine predators to ecological change [28,29]. Such indicators are particularly relevant for the Benguela which is considered to be critically located in terms of the global climate system and vulnerable to any future climate change or variability [30]. Indeed, changes in the Benguela have already been observed, with a turning point in the early 1990s leading to a warming of the waters on the west coast of South Africa and an increase in intra-annual upwelling variability [31]. Coincident with such changes have been marked alterations in the distribution and abundance of several forage species in the southern Benguela [32–34], which more or less coincides with the west coast of South Africa and the southern portion of Namibia's marine environment to the south of the Lüderitz area [35]. These include southward and eastward shifts in the distributions of several important prey species including sardine, anchovy and West Coast rock lobster. Correspondingly, there have been associated effects on the survival, abundance, distribution, feeding behaviour and diet of several top predator populations such as the African penguin (*Spheniscus demersus*), Cape gannet (*Morus capensis*) and Cape cormorant (*Phalacrocorax capensis*) in the southern Benguela [36–40]. Changes that have been documented include longer foraging trips, feeding on suboptimal food, reduced adult and chick survival, declines in abundance and eastward shifts in distribution.

Published information on the dive behaviour of Cape fur seals is currently available from only two individuals in 1977 [25], and there is some unpublished, preliminary data from 1994 to 1997 of the previous Sea Fisheries Research Institute (SFRI) of South Africa, which is now the Oceans and Coasts branch of the Department of the Environment, Forestry and Fisheries (DEFF). Consequently, there is a pressing need for information on the foraging behaviour of the Cape fur seal to provide a greater understanding of the potential consequences of future changes in its ecosystem. As adult females, otariid seals (fur seals and sea lions) must adopt a central place foraging strategy while provisioning pups, whereas other sex–age-classes are free to roam in search of optimal resources, therefore lactating females provide the best indication of local feeding conditions [41].

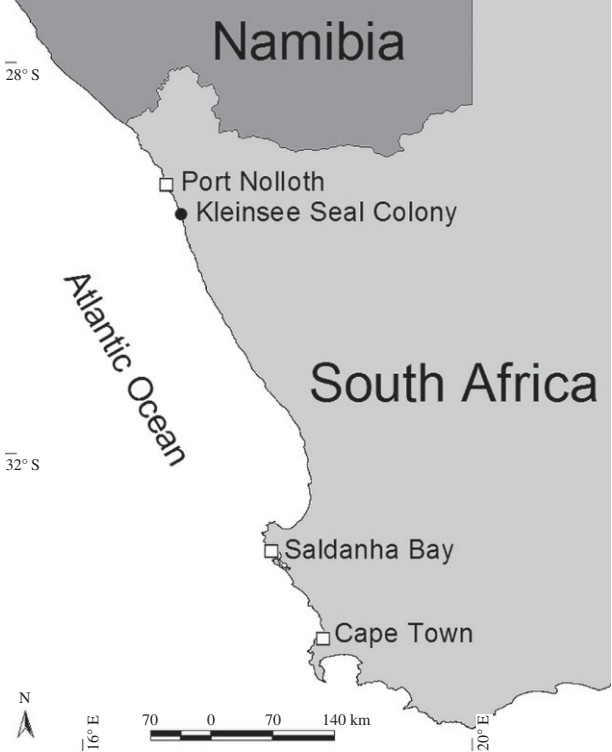

**Figure 1.** Location of the study site at the Kleinsee seal colony on the west coast of South Africa.

The aim of the present study, therefore, was to determine in adult female, Cape fur seals provisioning pups: (i) dive behaviour and foraging effort; (ii) inter-individual and inter-annual variation; and (iii) biological and physiological factors influencing these parameters.

# 2. Material and methods

## 2.1. Field data collection

The study was conducted at the Kleinsee seal colony on South Africa's west coast, approximately 100 km south of the border with Namibia (figure 1). The colony occurs on the mainland in a restricted-access strip of coastline that is zoned for diamond mining. Kleinsee was the largest Cape fur seal colony at the time of this study, with an average annual pup production of just over 80 000 pups estimated during censuses between 2000 and 2005 [2013].

Adult females nursing pups were selected at random and captured using a modified hoop net [42] during the austral winters of 2006–2008. Once restrained, anaesthesia was induced using isoflurane gas delivered via a portable vaporizer (Stinger, Advanced Anaesthesia Specialists, Gladesville, New South Wales, Australia; [43]). Anaesthetized individuals were removed from the capture net and weighed on a platform with a suspension scale (±0.5 kg) and measurements of standard length (SL), axillary girth (AG) and fore-flipper length (FFL) were taken using a metal tape measure (±0.5 cm). A dive recorder data logger (MK10-V, MK10-L or MK10-AF, Wildlife Computers, Redmond, WA, USA), programmed to record depth and temperature at 5 s intervals, was then glued to the pelage along the dorsal midline, just posterior to the scapula, using quick-setting epoxy (Araldite AW2101, CIBA-GEIGY Ltd). A VHF transmitter (Sirtrack, Ltd, Havelock North, New Zealand) was also attached posterior to the dive logger to facilitate re-location of animals on their return from foraging. As part of concurrent studies, some individuals received additional instruments, either a FastLoc GPS data logger (Sirtrack Ltd, Havelock North, New Zealand), a heart rate monitoring data logger or a stomach temperature data logger (Wildlife Computers, Redmond, WA, USA). Data for these additional devices are not reported here. In all cases, the attached devices in total represented less than 0.5% of body mass and less than 2% of cross-sectional surface area of the animal. Individually numbered plastic 'Jumbo' rototags (Dalton Supplies Ltd, Henley-on-Thames, UK) were inserted into the trailing edge of

each fore-flipper and paint marks were applied to the head and back to aid in locating animals for recapture. Following completion of instrumentation procedures (usually within 45 min of capture), individuals were allowed to recover from the anaesthetic and resume normal behaviour.

Upon their return to the colony after one or more foraging trips to sea, the study animals were located using a VHF receiver and visual aids (spotting scope, binoculars), and were captured as previously described. The data loggers were removed by cutting the fur beneath them with a scalpel before the animals were re-released. Data were downloaded from the loggers onto portable computers and later analysed in the laboratory.

For individuals sampled in 2006, additional information was obtained to estimate their body oxygen stores. Before anaesthesia induction, a 5 ml blood sample was collected by venepuncture from the caudal-gluteal vein for haematocrit (Hct) and haemoglobin (Hb) determination. In the field laboratory, Hct (%) was measured in triplicate from capillary tubes following centrifugation for 5 min at $12\,000g$ and a subsample (10 µl) of whole blood was also pipetted into 2.5 ml of Drabkin's reagent (Ricca Chemical Company, Arlington, TX, USA) before being stored in the dark until analysis. To determine plasma (Pv) and blood (Bv) volumes, a weighed (±0.1 g) intravenous dose of $ca$ 0.6 mg kg$^{-1}$ of Evans blue dye (Sigma Diagnostics, St Louis, MO, USA) was injected into the caudal-gluteal vein followed by the collection of three blood samples at 10, 20 and 30 min after the injection. Only injections where the injectate was completely introduced into the vein, determined by the ability to flush the injection syringe three times with blood after introduction of the dye, were used. The plasma fraction was separated and stored frozen (−20°C) until analysis. To determine muscle oxygen stores, a 6 mm dermal biopsy punch (Miltex, Inc., York, PA, USA) was used to obtain a muscle biopsy from a primary locomotor muscle (supraspinatus muscle) above the scapular and stored (−80°C) until analysis.

## 2.2. Data processing and analyses

Dive data were analysed using the *diveMove* package [44] in the R statistical environment (R v. 3.0.0; [45]). Zero offset correction of diving depth was performed to remove any artefacts arising from temporal changes in the accuracy of pressure transducers [46]. Following Page *et al.* [47], only dives greater than 4 m in depth were considered to be potential foraging dives because shallower dives are likely to be associated with travelling behaviour and/or be caused by waves. Dives and their phases (descent, bottom, ascent and post-dive phases) were identified and characterized in terms of duration and maximum depth achieved, while the shape of the dive was classified according to the proportion of time spent at depth and the variation in depth during that time [44]. The times of departure from and return to the colony were determined from the wet/dry sensor of the instruments, allowing trip duration to be calculated. Times are presented as South African Standard Time (SAST; = GMT + 2 h).

As a metric of dive effort during foraging trips, dive rate [48,49] was calculated as the rate of vertical distance travelled (dive rate, m h$^{-1}$) during the trip

$$DR = \frac{\sum (MDD*2)}{TD},$$

where DR is the dive rate, MDD is the maximum depth of dive and TD is the trip duration.

To investigate possible influences of year effects and individual morphology and dive type on proxies of foraging effort (trip duration and dive rate), linear models (LMs) were constructed in R. Response variables of trip duration (days) and dive rate (m h$^{-1}$) were modelled against predictor variables of year, body mass (kg), morphometric measurements (SL, AG and FFL, the ratio of FFL/SL) and body condition index (BCI; calculated from the residuals of the straight line regression between SL and body mass [50]. Also included as predictor effects were the proportion of benthic and nocturnal dives for each foraging trip. To assess the factors influencing benthic diving, the proportion of benthic dives per trip were modelled in response to year and the same morphological data as above, as well as proportion of nocturnal dives for each foraging trip.

The initial inspection of predictor effects revealed that body mass was highly correlated with both SL and AG measurements (Pearson's correlation test, $r > 0.7$) and was thus excluded from the model to reduce collinearity between predictor effects. In addition, measurements for FFL and the ratio of FFL/ SL were absent for three of the sampled individuals. Log-likelihood ratio tests (LRTs) on a subset of the data revealed that neither FFL nor the FFL/SL ratios had a significant effect on the response variables (all $p > 0.1$). As such, both of these variables were excluded from further analysis. Model selection was undertaken using Akaike information criterion scores corrected for small sample sizes (AICc), calculated for all permutations of the predictor effects using the dredge function in the

R package *MuMIn* [51]. A subset of the most parsimonious models was identified as those with a ΔAIC of less than or equal to 4 and used to generate model averaged coefficient estimates [52,53].

To determine whether distinctive foraging strategies might be exhibited within the sampled seals, a hierarchical agglomerative clustering [54] was then performed using PRIMER 6 (Plymouth Routines In Multivariate Ecological Research, [55,56]), on the resemblance matrix of the Euclidean distances between samples (seals). Variables included trip duration (days), modal dive duration (s), modal dive depth (m), dive rate (m h$^{-1}$), proportion of dives in a trip that were conducted at night and proportion of dives that were benthic. Data were standardized by the total of each variable to balance the contributions of variables measured on different scales. The year of study was included as a factor.

The behavioural aerobic dive limit (bADL) of each animal was estimated from the relationship between dive duration and post-dive interval using the intersection of first percentile quantile regressions [57] produced using the *quantreg* package [58] in R. Factors influencing the proportion of dives that exceeded the bADL were investigated using LMs. The proportion of dives that exceeded the bADL for each individual were modelled in response to morphometric measurements (SL, AG), BCI, dive rate and the overall proportion of benthic dives. As above, model selection (AICc) and model averaging were then undertaken.

For individuals sampled in 2006, total body oxygen stores were determined following the methods of Fowler *et al.* [59], by adding the blood, muscle and lung oxygen stores [60–65]. Lung oxygen stores were derived from allometric estimates of lung volume for otariids following Costa *et al.* [65] while blood and muscle oxygen stores were obtained in this study. Blood Hb concentration was determined in the laboratory by the cyanmethaemoglobin photometric method (Stanbio Laboratory, Boerne, TX, USA), and mean corpuscular haemoglobin content (MCHC) was calculated using the equation: MCHC = Hb/Hct. Plasma volume was determined by photometric absorbance of the plasma samples at 624 and 740 nm and the calculation of the dye concentration at the time of injection following the Evans blue method [66–68], and blood volume (Bv) was calculated from Pv and Hct. Myoglobin (Mb) concentration in the muscle samples was determined following the method of Reynefarje [69]. By adding the blood, muscle and lung oxygen stores, the total body oxygen store was obtained [60–65].

Unless otherwise stated, data are presented as mean ± s.e.

## 3. Results

Foraging trip dive data were obtained from a total of 32 adult female Cape fur seals (10 in 2006, 9 in 2007 and 13 in 2008). Body mass, SL, AG and FFL at capture were 53.8 ± 1.4 kg, 135.8 ± 1.4 cm, 90.0 ± 1.1 cm, 42.0 ± 0.5 cm, respectively (table 1). Individuals were recaptured after one to three foraging trips, but records of complete trips were available only for the first trip in most animals. Hence, to standardize analyses, only the first foraging trip post-release was investigated; all reported results refer only to these first trips. Trip duration ranged from 1.75 to 14.46 days (5.40 ± 0.21 days) (table 1), with no significant difference between the years of sampling (ANOVA, $F_{2,29} = 0.305$, $p > 0.7$). Individuals departed the colony for their foraging trip mostly at night between 0.00 and 06.00 and also late morning between 10.00 and 12.00 (electronic supplementary material, figure S1), with the mean time at sea before the first foraging dives being 3.7 ± 0.4 h. Arrival back at the colony was mainly just after dusk at 18.00–20.00.

A total of 37 024 dives were recorded (161–2317 dives per animal). Dive depths and durations were highly skewed within each individual (figure 2), and therefore, modes, medians and maximums are reported here: the modal dive depth per animal ranged from 4.3 to 196.0 m; the mean modal depth across animals was 42.1 m ± 11.8 and the deepest depth recorded for any animal was 454 m. The median dive depth ranged from 14.0 to 176.5 m; the mean of medians was 59.6 m ± 7.8. The modal dive duration per animal ranged from 0.2 to 5.6 min; the mean modal dive duration across animals was 2.0 min ± 15.8 and the longest dive recorded for any animal was 9.8 min. The median dive durations ranged from 1.3 to 5.3 min (mean of the medians = 2.3 min ± 0.5). The dive rate over complete foraging trips averaged 1127 ± 47.2 m h$^{-1}$. Dive and trip statistics per animal (mode, median and mean) are shown in electronic supplementary material, table S1.

Dive profiles could be grouped into four main categories (figure 3a). The most common dive type was pelagic U-shaped (53.1 ± 3.0%), followed by V-shaped pelagic dives (13.4 ± 1.3%). A third type of pelagic dive (17.1 ± 2.9%) was observed in which individuals entered what appeared to be the bottom phase of a U-shaped dive at a particular depth, but after beginning to ascend, changed direction and descended to much greater depths (mean = 97.0 m ± 0.4). The second descent period had faster descent rates as indicated by the slopes of the dive profile. The remainder of the dives (16.4 ± 3.2%) were flat-bottom

**6**

**Table 1.** Morphometric measurements and derived indices, trip duration and dive effort recorded for the 32 study animals from the Kleinsee colony during the 2006–2008 study period. SL represents the standard length, AG is axillary girth, FFL is fore-flipper length, BCI is body condition index and was calculated as the residuals of the linear regression between SL and mass.

| seal | SL (cm) | mass (kg) | AG (cm) | FFL (cm) | FFL/SL | BCI | trip start date | trip duration (days) | dives (n) | dive rate (m h$^{-1}$) | benthic dives (%) |
|---|---|---|---|---|---|---|---|---|---|---|---|
| 1 | 132 | 48.8 | 92 | 40 | 0.30 | 2.18 | 2 July 2006 | 2.6 | 762 | 728 | 0 |
| 2 | 129 | 45 | 90 | | | 3.7 | 5 July 2006 | 5.2 | 1406 | 1086 | 1 |
| 3 | 136 | 47.5 | 85 | 38 | 0.28 | 6.52 | 6 July 2006 | 8.5 | 1536 | 1263 | 3 |
| 4 | 143 | 53 | 89 | 44 | 0.31 | 6.34 | 6 July 2006 | 2.1 | 460 | 1162 | 22 |
| 5 | 139 | 57.5 | 102 | | | −1.2 | 4 July 2006 | 1.7 | 161 | 827 | 24 |
| 6 | 136 | 54.5 | 94 | 49 | 0.36 | −0.48 | 6 July 2006 | 7.3 | 1480 | 1000 | 0 |
| 7 | 131 | 59 | 99 | 44 | 0.34 | −8.78 | 7 July 2006 | 10.7 | 1289 | 1500 | 8 |
| 8 | 143 | 58 | 97 | | | 1.34 | 5 July 2006 | 9.1 | 1478 | 913 | 15 |
| 9 | 110 | 34.5 | 83 | 40 | 0.36 | −0.25 | 6 July 2006 | 4.1 | 1588 | 992 | 6 |
| 10 | 137 | 52.5 | 89 | 47 | 0.34 | 2.28 | 5 July 2006 | 8.7 | 2196 | 1035 | 0 |
| 11 | 125 | 51.8 | 92 | 40 | 0.32 | −6.14 | 23 June 2007 | 4 | 908 | 1171 | 18 |
| 12 | 140 | 59.6 | 95 | 40 | 0.29 | −2.54 | 25 June 2007 | 3.2 | 675 | 1360 | 14 |
| 13 | 138 | 52.4 | 92 | 42 | 0.30 | 3.14 | 25 June 2007 | 5.7 | 755 | 1012 | 41 |
| 14 | 125 | 47.6 | 84 | 43 | 0.34 | −1.94 | 26 June 2007 | 5.1 | 1288 | 1554 | 9 |
| 15 | 144 | 54 | 87 | 45 | 0.31 | 6.1 | 25 June 2007 | 4.4 | 843 | 1012 | 1 |
| 16 | 146 | 78.4 | 108 | 46 | 0.31 | −16.8 | 28 June 2007 | 4.9 | 1677 | 1554 | 11 |
| 17 | 137 | 53.2 | 89 | 44 | 0.32 | 1.58 | 27 June 2007 | 7.1 | 1366 | 1208 | 0 |
| 18 | 135 | 54.4 | 92 | 42 | 0.31 | −1.14 | 28 June 2007 | 8.9 | 335 | 1042 | 19 |
| 19 | 140 | 60 | 97 | 42 | 0.30 | −2.94 | 27 June 2007 | 2.1 | 1095 | 1662 | 80 |
| 20 | 148 | 58.6 | 94 | 46 | 0.31 | 4.16 | 24 June 2008 | 5.4 | 445 | 1014 | 10 |
| 21 | 130 | 46.2 | 82 | 40 | 0.31 | 3.26 | 21 June 2008 | 2 | 991 | 923 | 19 |

(Continued.)

**Table 1.** (*Continued.*)

| seal | SL (cm) | mass (kg) | AG (cm) | FFL (cm) | FFL/SL | BCI | trip start date | trip duration (days) | dives (n) | dive rate (m h$^{-1}$) | benthic dives (%) |
|---|---|---|---|---|---|---|---|---|---|---|---|
| 22 | 135 | 49.2 | 85 | 41 | 0.30 | 3.68 | 20 June 2008 | 5.2 | 592 | 1185 | 5 |
| 23 | 125 | 41.6 | 83 | 38 | 0.30 | 3.68 | 21 June 2008 | 4 | 1448 | 568 | 15 |
| 24 | 152 | 68.4 | 94 | 43 | 0.28 | −2.6 | 21 June 2008 | 14.5 | 2317 | 865 | 7 |
| 25 | 136 | 53.8 | 89 | 40 | 0.29 | −0.16 | 24 June 2008 | 7.8 | 547 | 1306 | 6 |
| 26 | 137 | 63.4 | 90 | 43 | 0.31 | −8.62 | 23 June 2008 | 2.7 | 545 | 1453 | 34 |
| 27 | 135 | 59 | 80 | 39 | 0.29 | −6.12 | 20 June 2008 | 3.8 | 1288 | 1230 | 43 |
| 28 | 137 | 56.2 | 88 | 41 | 0.29 | −1.42 | 22 June 2008 | 3.5 | 333 | 947 | 56 |
| 29 | 130 | 49 | 84 | 40 | 0.30 | 0.08 | 23 June 2008 | 6.3 | 1647 | 1019 | 2 |
| 30 | 134 | 50.2 | 85 | 38 | 0.28 | 2.3 | 23 June 2008 | 4.6 | 795 | 742 | 4 |
| 31 | 139 | 53.4 | 89 | 43 | 0.31 | 2.9 | 23 June 2008 | 3.2 | 556 | 962 | 32 |
| 32 | 140 | 49.2 | 81 | 41 | 0.29 | 7.86 | 22 June 2008 | 4.5 | 1035 | 1630 | 20 |
| mean | 135.8 | 53.7 | 90.0 | 42.0 | 0.31 | | | 5.4 | 1057.4 | 1122.7 | 16.4 |
| s.e. | 1.4 | 1.4 | 1.1 | 0.5 | 0.004 | | | 0.5 | 94.5 | 16.5 | 4.3 |

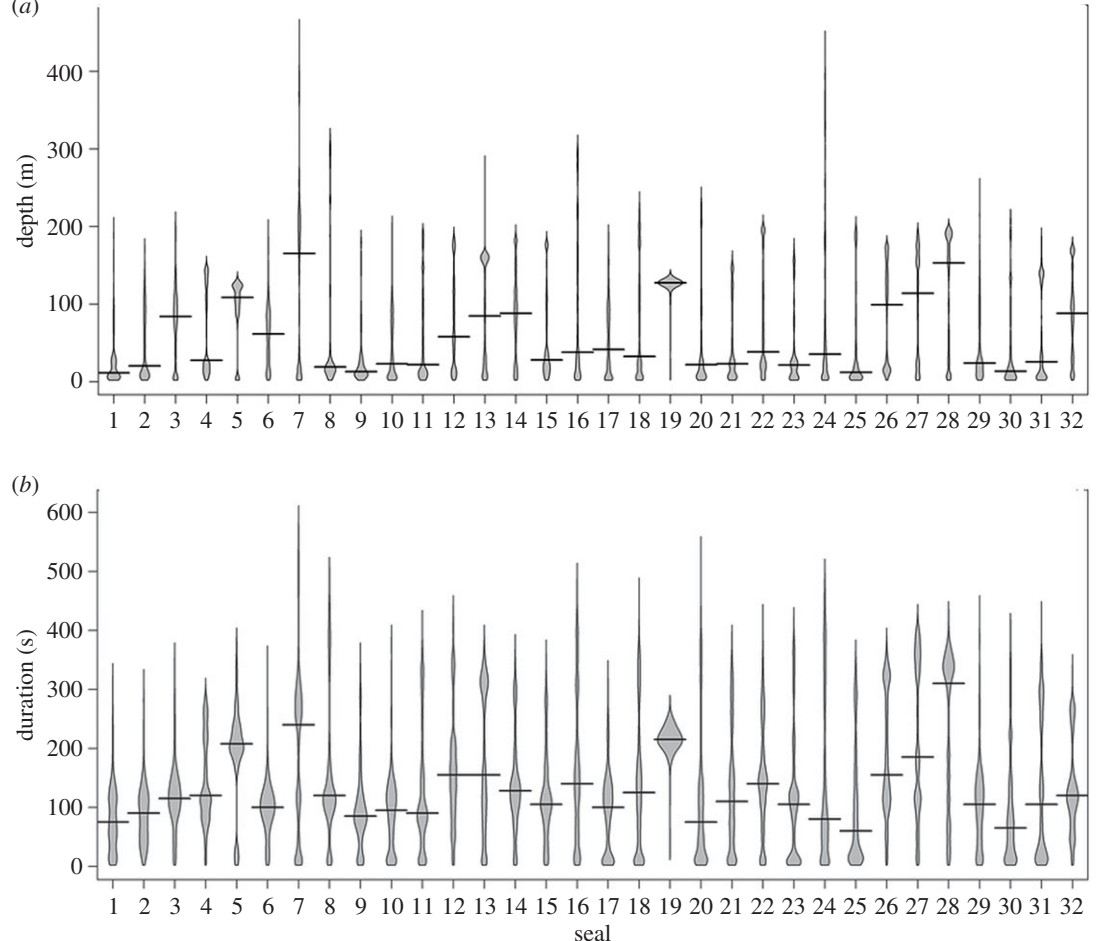

**Figure 2.** Beanplots summarizing (*a*) the frequencies of maximum depths attained per dive by each of 32 lactating Cape fur seal females fitted with depth recorders at the Kleinsee seal colony, South Africa, during the austral winters of 2006–2008; and (*b*) the frequencies of dive durations by each of the 32 females. The beanlines (solid) represent the median dive depth or duration per female.

in profile, with consistent maximum depths between sequential dives, indicative of benthic foraging [49]. Maximum depths attained during benthic dives (mode of all animals = 175.0 m) were most frequently greater than those during other dive types ('pursuit': mode = 98.0 m, U-shaped: mode = 9.0 m, V-shaped: mode = 4.5 m; figure 3*b*). However, the greatest depths were recorded during U- and V-shaped pelagic dives (max = 454.0 m). The proportions of benthic and 'pursuit' dives were variable among females, with both shapes being either absent or negligible in some animals or abundant (greater than 50%) in others (figure 3*c*).

There was diel variation in dive depth, the greatest depths being during daylight hours (figure 4*a*). However, this period coincided with the lowest frequency of dives, whereas the greatest intensity of diving occurred at night, peaking at dusk and dawn (figure 4*b*); this was true of the three pelagic dive types while benthic diving took place almost exclusively during daylight hours (figure 4*b*).

Candidate subsets of the most parsimonious LMs (ΔAICc ≤ 4) explaining trip duration, dive rate and the proportion of benthic diving, comprised a total of 8, 20 and 5 models, respectively. The most likely set of models for trip duration retained all morphometric measurements, as well as the proportion of benthic and nocturnal dives as predictor effects (electronic supplementary material, table S2). However, model averaging indicated that the strongest predictor of trip duration was the proportion of benthic diving (95% confidence interval did not cross zero), revealing an inverse relationship between these two parameters (table 2). While candidate models for dive rate retained all predictor effects (electronic supplementary material, table S2), model averaging revealed no clear effect of any of the predictor variables (table 2). The most likely candidate models for the proportion of benthic diving retained all morphometric measurements and the proportion of nocturnal dives as predictor effects (electronic supplementary material, table S2). Model averaging, in turn, indicated that only the proportion of

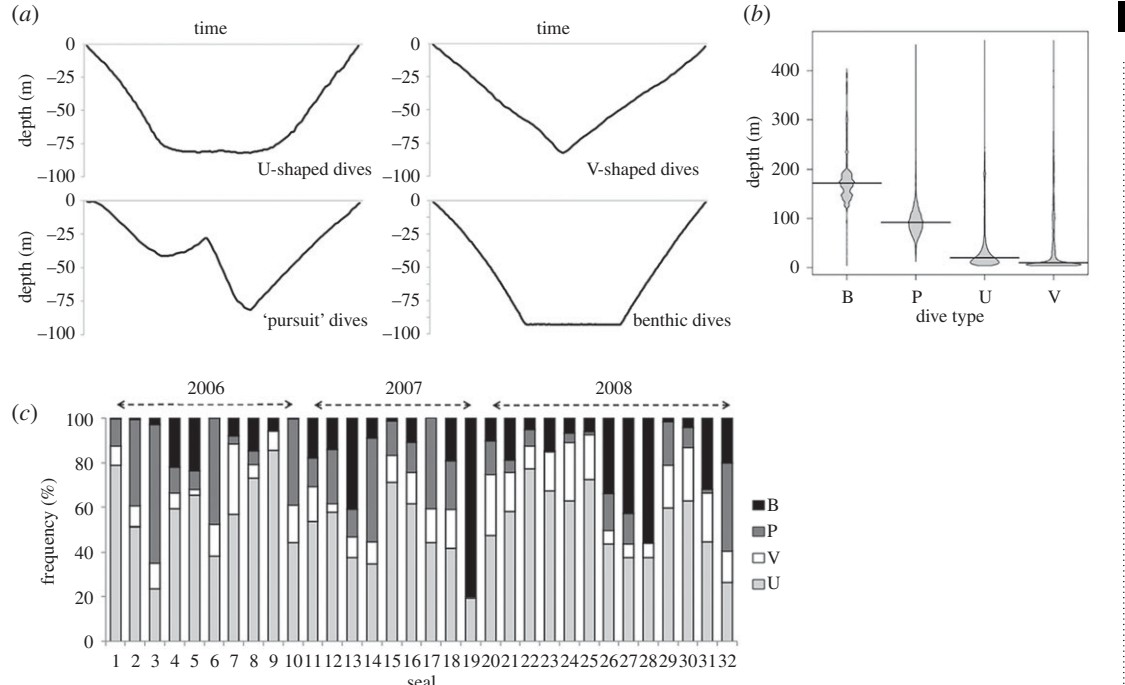

**Figure 3.** (a) Examples of dive profiles of the four dive types identified for Cape fur seal females fitted with depth recorders at the Kleinsee seal colony, South Africa, during the austral winters of 2006–2008, namely U-shaped dives (U), V-shaped dives (V), 'pursuit' dives (P) and flat-bottomed benthic dives (B); (b) beanplots summarizing the frequencies of maximum depths for the four dive types (the beanlines represent the median dive depth per dive type); (c) the percentage frequencies of the different dive types identified for each female.

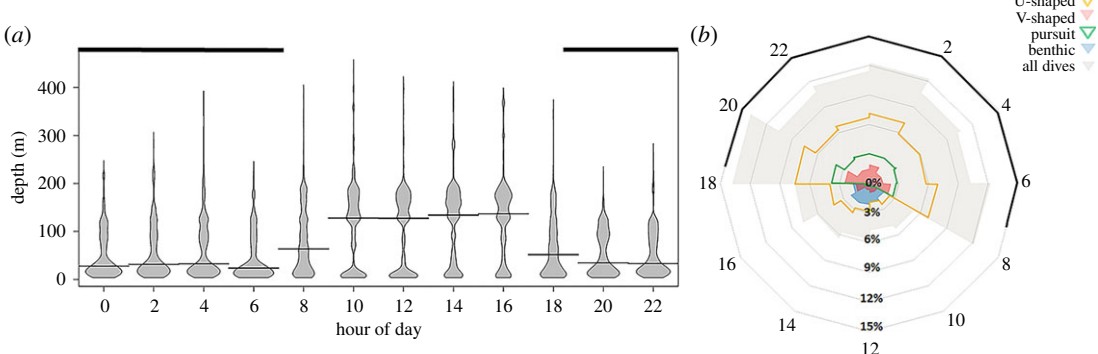

**Figure 4.** (a) Beanplots summarizing the frequencies of maximum dive depths attained per each hour of day (SAST) by all 32 lactating Cape fur seal females fitted with depth recorders at the Kleinsee seal colony, South Africa, during the austral winters of 2006–2008. The x-axis represents hours of the day in 2 h bins (i.e. summarizing the dives for the preceding 2 h), the beanlines represent the median value, and the shaded parts of the x-axis represent the time between dusk and dawn (mean civil twilight times coinciding with the dates of the study for the nearby town of Port Nolloth, South Africa). (b) Radial graphs summarizing the frequencies of dives of different dive types attained per time of day by each animal. The vertical axis labels represent the percentage frequency of all dives, the radial axis labels represent the time of day, with the dives for the preceding 2 h summarized in each bin, and the dark shading on the outer radial line represents the hours between dusk and dawn.

nocturnal dives had a substantial effect on benthic diving, with a strong inverse relationship between these two parameters (table 2).

The hierarchical clustering analysis separated the sampled individuals into two main groups according to the characteristics of their foraging trips and diving behaviour (figure 5a); each cluster contained individuals from all three years. The smaller cluster, consisting of only seven animals, had significantly longer dive durations (figure 5c) and dive depths (figure 5d) than the larger cluster ($n = 25$), and significantly less of their dives were nocturnal (figure 5f) while significantly more of their dives were benthic (figure 5g). There were no significant differences in trip duration or dive rate

**Table 2.** Model averaged coefficient estimates (CE) for the factors influencing foraging trip duration, dive rate, the proportion of benthic dives and the proportion of dives that exceeded bADL for female Cape fur seals at Kleinsee, South Africa. Italicized parameter estimates indicate those for which the 95% confidence intervals did not include zero.

| response variables | predictor effect | CE ± s.e. | z-value (p) | relative importance | n models |
|---|---|---|---|---|---|
| foraging trip duration | intercept | −3.16 ± 9.76 | 0.32 (0.75) | | |
| | *proportion of benthic dives* | *−0.07 ± 0.03* | *2.50 (0.01)* | *1.00* | *8* |
| | standard length | 0.06 ± 0.07 | 0.87 (0.38) | 0.58 | 4 |
| | body condition index | −0.02 ± 0.06 | 0.34 (0.73) | 0.21 | 2 |
| | axillary girth | 0.01 ± 0.04 | 0.27 (0.79) | 0.18 | 2 |
| | proportion of nocturnal dives | 0.001 ± 0.02 | 0.05 (0.96) | 0.13 | 2 |
| dive rate | intercept | 1053.54 ± 543.82 | 1.88 (0.060) | | |
| | year$^{2007}$ | 119.23 ± 148.12 | 0.79 (0.43) | 0.51 | 9 |
| | year$^{2008}$ | −4.69 ± 95.41 | 0.05 (0.96) | | |
| | body condition index | −5.68 ± 10.21 | 0.55 (0.58) | 0.36 | 8 |
| | proportion of benthic dives | 0.81 ± 2.06 | 0.39 (0.69) | 0.23 | 5 |
| | axillary girth | −1.81 ± 7.76 | 0.23 (0.82) | 0.17 | 5 |
| | standard length | 1.15 ± 4.37 | 0.26 (0.79) | 0.17 | 5 |
| | proportion of nocturnal dives | 0.61 ± 2.58 | 0.23 (0.82) | 0.16 | 4 |
| proportion of benthic dives | intercept | 76.33 ± 32.77 | 2.24 (0.03) | | |
| | *proportion of nocturnal dives* | *−0.94 ± 0.23* | *3.92 (<0.001)* | *1.00* | *5* |
| | body condition index | −0.17 ± 0.39 | 0.42 (0.68) | 0.29 | 2 |
| | standard length | −0.03 ± 0.15 | 0.17 (0.87) | 0.19 | 2 |
| | axillary girth | −0.02 ± 0.21 | 0.08 (0.94) | 0.14 | 1 |
| proportion of dives exceeding bADL | intercept | 0.08 ± 0.17 | 0.46 (0.64) | | |
| | *proportion of benthic dives* | *0.01 ± 0.001* | *6.62 (<0.001)* | *1.00* | *7* |
| | axillary girth | −0.001 ± 0.002 | 0.97 (0.33) | 0.25 | 3 |
| | standard length | 0.001 ± 0.001 | 1.01 (0.31) | 0.24 | 3 |
| | dive rate | 0.000003 ± 0.00002 | 0.55 (0.59) | 0.15 | 2 |
| | body condition index | −0.001 ± 0.003 | 0.53 (0.59) | 0.10 | 1 |

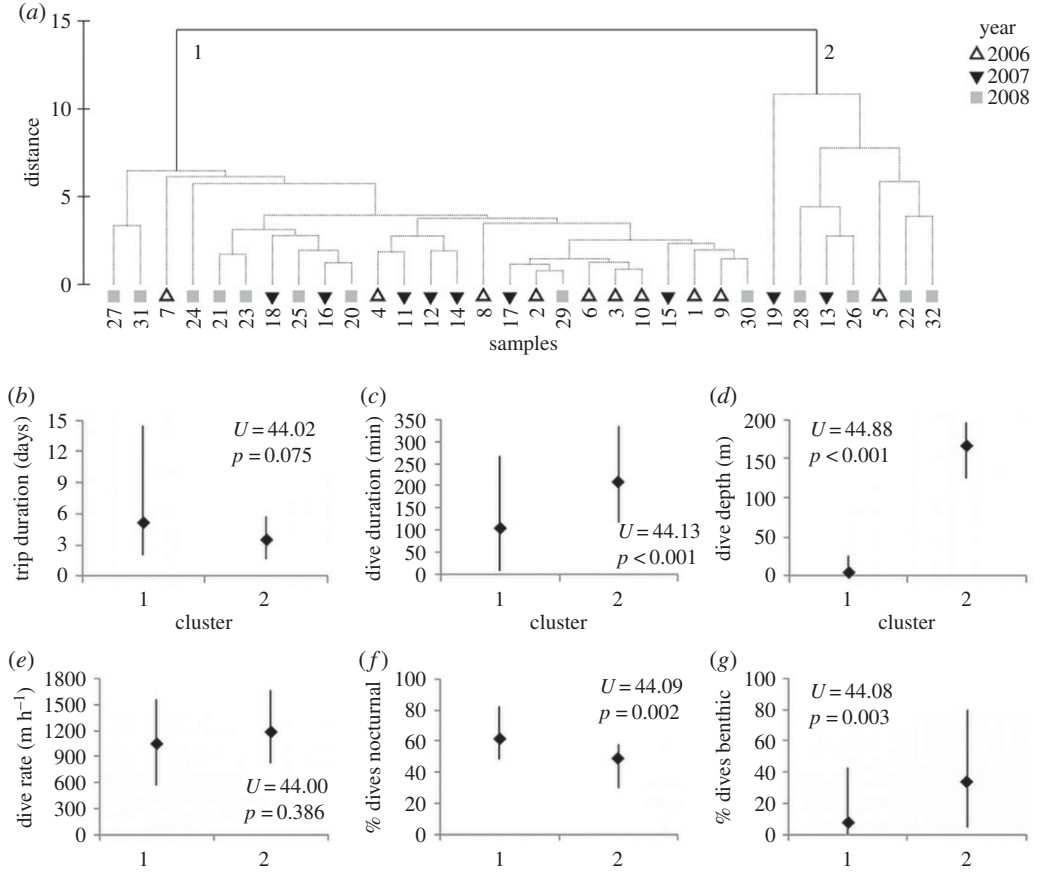

**Figure 5.** Cluster analysis plot (a) of the diving and foraging trip characteristics of individual study animals (x-axis labels = seal identity). The variables, which included trip duration (days), modal dive duration (s), modal dive depth (m), dive rate (m h$^{-1}$), % of dives in a trip that were conducted at night and % of dives that were benthic, were standardized by the total of each variable. Hierarchical agglomerative clustering was performed on the resemblance matrix of the Euclidean distances between samples (seals), with year of study as factor. Also shown (b–g) are comparisons of the input variables between the two main clusters from (a). The markers represent the median value and the error bars the range of values; U represents the critical U-value for the one-tailed Mann–Whitney test for two independent samples, with p-value.

between the clusters (Mann–Whitney tests, $p > 0.5$ in both cases). There were no significant differences between the two clusters in the SL, body mass, AG, FFL, the ratio between fore-flipper and SL or BCI of the individuals (Mann–Whitney tests, $p > 0.5$ in all cases).

There was a sufficient number of dives with clearly extended post-dive intervals in 18 individuals to enable the calculation of bADL ($231 \pm 10$ s). The proportion of dives that exceeded the bADL varied between these individuals (1.5–41.5%). Most dives that exceeded bADL were benthic, although this too varied among individuals (figure 6). Indeed, while subsets of the most likely LMs retained all predictor effects (electronic supplementary material, table S2), model averaging indicated that only the overall proportion of benthic dives had a strong influence on the proportion of dives that exceeded the bADL (table 2).

Data on total oxygen stores were obtained from 9 of the 10 individuals sampled in 2006 (table 3). Mb ($4.6 \pm 0.1$ g 100 g$^{-1}$ wet tissue), Hct ($55.2 \pm 0.7$%), Hb ($16.3 \pm 0.2$ g dl$^{-1}$) and total oxygen stores ($43.3 \pm 1.3$ mlO$_2$ kg$^{-1}$) varied little between individuals, while plasma volume ($35.6 \pm 3.0$ ml kg$^{-1}$) and blood volume ($79.6 \pm 6.6$ ml kg$^{-1}$) were more variable. However, there were no correlations between either of the latter two variables and modal dive depth or modal dive duration of individuals (Spearman's rank correlation coefficient, $p > 0.05$ in all cases).

## 4. Discussion

Knowledge of the foraging behaviour of top predators is crucial for understanding their ecosystem function and predicting how they may respond to environmental variability [70]. The present study

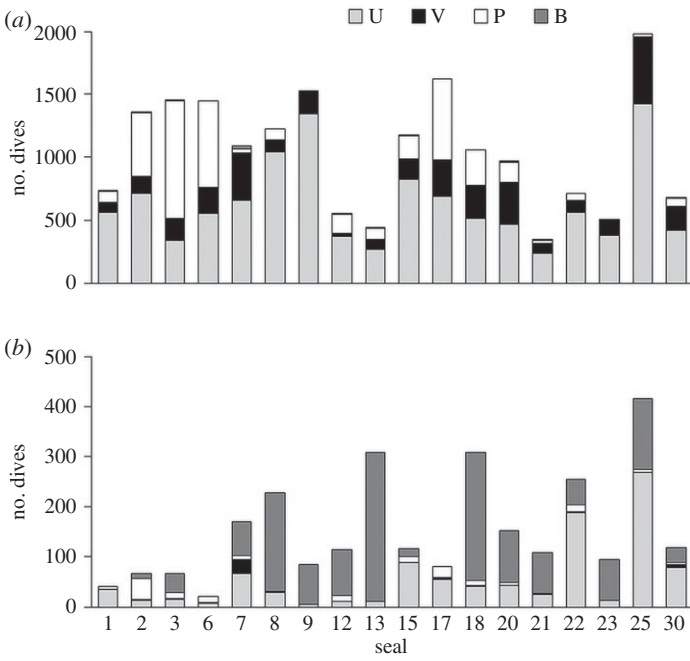

**Figure 6.** Numbers of dives with durations that were below (*a*) and above (*b*) the bADL in 18 Cape fur seal females. Dive types are indicated: U- and V-shaped pelagic, P 'pursuit' pelagic and B benthic.

**Table 3.** Summary of body oxygen store components in female Cape fur seals from Kleinsee, South Africa, in 2006. Values for plasma volume (Pv), blood volume (Bv), haematocrit (Hct), haemoglobin (Hb) and myoglobin (Mb) in wet tissue are given.

| seal | Pv (ml kg$^{-1}$) | Bv (ml kg$^{-1}$) | Hct (%) | Hb (g dl$^{-1}$) | Mb (g 100 g$^{-1}$) | O$_2$ stores (ml kg$^{-1}$) |
|---|---|---|---|---|---|---|
| 1 | 36 | 85 | 58 | 17.1 | 4.91 | 46.5 |
| 2 | 34 | 71 | 52 | 14.95 | 4.18 | 38.7 |
| 3 | 48 | 104 | 54 | 16.26 | 4.58 | 47.4 |
| 4 | 36 | 88 | 59 | 16.78 | 4.45 | 44.7 |
| 5 | 32 | 72 | 56 | 16.8 | 4.66 | 42.8 |
| 6 | 17 | 37 | 55 | 16.27 | 4.64 | 36.3 |
| 7 | 35 | 78 | 55 | 16.27 | 4.61 | 43.0 |
| 9 | 47 | 102 | 54 | 16.12 | 4.79 | 47.8 |
| 10 | 36 | 79 | 54 | 16.15 | 4.44 | 42.3 |
| mean | 35.7 | 79.6 | 55.2 | 16.3 | 4.6 | 43.3 |
| s.e. | 3.0 | 6.6 | 0.7 | 0.2 | 0.1 | 1.3 |

investigated the foraging behaviour of pup-rearing Cape fur seal females, the most ecologically important component of the population for the greatest marine top predator biomass species in southern Africa (greater than seabirds, cetaceans and apex shark predators; [16]). The results reveal that the majority of dives were pelagic and to depths of less than 100 m, indicating that the Cape fur seal at Kleinsee is primarily an epipelagic forager, although a small proportion of individuals conducted a substantial number of benthic dives. Individuals displayed diel variation, diving to shallower depths at night and reaching the deepest recorded dives for females of any fur seal species during daylight hours (*ca* 450 m). These findings are consistent with previously published reports of the Cape fur seal diet from the Benguela comprising primarily schooling fish [18,22,71] and highlight the potential negative consequences of environmental changes affecting the distribution of such prey [72] on foraging effort for the population.

## 4.1. Intra- and inter-individual variation in dive behaviour

The timing of departure from the colony has been found in previous studies of fur seals to reflect the time taken to arrive at the nearest foraging areas at the most profitable period of day [47,73]. In the present study, the majority of individuals departed the colony at night and did not commence foraging until approximately 4 h later. As diving activity in the present study occurred primarily at night, it is not apparent that departure times conferred any temporal foraging benefit. This is in contrast with other diel diving species where departure from the colony appears to be timed so as to reach the foraging areas around sunset (e.g. [47]). The threat of predation in the vicinity of the colony such as by the white shark (*Charcaradon carcarias*), the most conspicuous predator of Cape fur seals in South Africa [74–76], could also influence movements to and from the colony, with departure at night reducing potential risks [77].

In the present study, while all individuals displayed a wide range of dive depths, modal dive depths per animal were mostly less than 20 m. This is reflective of the primarily epipelagic foraging mode of the Cape fur seal and comparable to other epipelagic foraging species consuming small schooling prey (e.g. Guadalupe fur seal *A. townsendi*, [78]; New Zealand fur seal *A. forsteri*, [79]). Similar to other epipelagic foraging fur seal species, individuals in the present study dived mainly at night to shallow depths. This is consistent with the diel vertical migrations observed in several important prey species of the Cape fur seals. For example, anchovy [80], sardine [81,82], round herring (*Etrumeus whiteheadi*) [83], horse mackerel [84] and juvenile hake [85] schools are typically found scattered above 50 m at night but deeper during the day.

Punt *et al.* [17] referred to unpublished dive data (SFRI 1994, unpublished data) for only two Cape fur seals (one male and one female), which indicated that there was less diving at night than at other times of the day. This they found contradictory to diet data from offshore feeding seals which showed that deep water hake (*Merluccius paradoxus*) was found in seal stomachs. It was expected that this species could only be caught at night by seals during their diel vertical migration. However, the lack of night diving in the dive data they reported led the authors to consider that deep water hake in the seal stomachs must have come from animals scavenging from trawl catches of this commercially important species. By comparison, the present study (based on 32 animals) demonstrated a prevalence of diving at night, illustrating the risk of basing such suppositions upon limited sample sizes and supports the notion that Cape fur seals could indeed be hunting deep water hake.

Most individuals had maximum dive depths of between 150 and 250 m and, while dive depths were generally deeper during daylight, such maximum depths were attained at all hours. Several individuals displayed much greater maximum dive depths (greater than 300 m), with two individuals reaching depths of 430–454 m. These are the deepest recorded dive depths for any female fur seal, with the maximum recorded for similar sized species being 312 m by the New Zealand fur seal [47]. By contrast, the maximum recorded female dive depth for the Australian fur seal, the largest fur seal species (mass 76 kg; [86]) and a conspecific of the Cape fur seal, is 164 m [87]. However, the Australian fur seal is almost exclusively a benthic/demersal forager [49,87,88] and the foraging range of adult females is restricted mostly to a shallow continental shelf (Bass Strait) where the average seafloor bathymetry is 60–80 m [87]. Hence, dive depth for this subspecies may not be constrained by physiological limits.

The wide range in dive durations (individual modal dive duration 0.2–5.6 min; longest recorded dive 9.8 min) observed in the present study is consistent with the high intra-individual variation that was observed in dive depths. The mean median dive duration (2.3 min) was greater than in the smaller subantarctic (*A. tropicalis*, 1.58 min) and Antarctic (*A. gazella*, 1.24 min) fur seals [89], comparable to the similar sized New Zealand fur seals (2.1 min, [47]) and lower than in the larger Australian fur seal (3.1 min, [87]).

Comparable to other epipelagic foraging fur seals (e.g. [47,90]), the predominant dive profile in the present study was pelagic U-shaped, interspersed occasionally with V-shaped dives. The similar depth of these two dive profiles when they co-occurred suggests foraging on schooling prey with the latter profile reflecting either a more rapid prey encounter or the capture of single larger prey. The occurrence of 'pursuit' dives, the second most common dive type in the present study and a profile not previously reported in seals, could also indicate the capture of single larger prey. These dives were characterized by an apparent first nadir, at a depth layer consistent with the maximum depths of the U- and V-shaped dives they were interspersed with, before individuals ventured substantially deeper into the water column with an increased descent rate, suggestive of active prey chase. The increased effort implied by the steeper descent rate, and the substantially greater depths attained during this phase, suggests individuals may opportunistically have encountered larger prey while at

the small schooling prey depth layer. This could be larger, piscivorous fish such as snoek (*Thyrsites atun*), which are highly mobile, opportunistic predators of small pelagic shoaling fish such as sardine and anchovy, and also of horse mackerel and lanternfish (family Myctophidae) [91,92], all of which are known to be preyed upon by Cape fur seals [18,20]. Future studies using animal-borne cameras are needed to confirm this hypothesis [93].

The limited previously available information on the diving behaviour ([25]; SFRI, unpublished data for a few individuals in 1994–1997), and extensive diet studies (e.g. [18,20,71]), of the Cape fur seal have suggested that it is primarily an epipelagic forager. This is consistent with its distribution being largely associated with the Benguela, an area characterized by very high marine primary productivity supporting a large biomass of small pelagic prey species [35,94,95] conducive to mid-water foraging by air-breathing marine vertebrates [96]. However, in the present study, benthic dives were observed in 90% of seals, comprising more than 50% of dives in two individuals. The majority of benthic dives were to depths of 120–200 m suggesting individuals foraged on demersal/benthic prey such as bearded gobies (*Sufflogobius bibarbatus*) or possibly cephalopods of the family Ommastrephidae, both known to occur in the Cape fur seal diet [20,97,98]. Individuals also made pelagic dives to beneath the depths where deeper-ranging epipelagic prey species may extend to (i.e. greater than 200 m; [83]), where they may have targeted mesopelagic species such as lanternfish.

Both gobies and lanternfish were found to be prominent in the previously sardine-dominated diet of Cape fur seals in Namibia following the 1970s stock collapse of sardines in the northern Benguela [20,71,92]. Therefore, the observed occurrence of benthic and mesopelagic foraging in the present study could reflect inadequate pelagic prey availability [96]. This is consistent with well-documented shifts in the distributions of spawning stocks of sardine and anchovy in South Africa towards the east Agulhas Bank since the 1990s [32,33]. These changes have resulted in decreased availability of these prey species in the southern Benguela for not only seals, but also seabirds such as the African penguin, Cape gannet and Cape cormorant, that are dependent on these prey [36–40].

All seals in the present study displayed multiple dive types and a wide range of dive depths, indicating a high degree of flexibility in foraging behaviour. This is consistent with previous diet studies of the Cape fur seal revealing consumption of a wide range of prey types [18,20,71,98,99]. Within this variability, two main foraging behaviour groupings were observed, with a small cluster characterized by longer dive durations and deeper dive depths than the larger cluster which conducted more daytime benthic dives. Whereas in some other otariids, multiple foraging strategies within populations have been associated with differences in body mass or morphometric differences (e.g. [100,101]), no individual body differences could be detected between the two foraging behaviour groups observed in the present study. However, further investigations are required spanning multiple foraging trips per animal (e.g. [101–103]) to more properly assess the presence of specific foraging strategies and individual factors influencing them.

## 4.2. Foraging effort

The mean foraging trip duration observed in the present study (5 days) was within the range previously reported for the Cape fur seal in summer at the same location (3–5 days, [104,105]) and in winter at Namibian colonies (5–6 days, [106]), and for other temperate fur seal populations during winter [47,87]. Foraging trip duration in female otariid seals has previously been considered as indicative of prey resource availability within the maternal foraging range from the breeding colony [48,73]. In the present study, there was no significant inter-annual difference in trip duration. Similarly, no effect of year of study was found on dive rate (vertical distance covered, m h$^{-1}$, an index of foraging effort, [48,73]) or the proportion of dives that were benthic or at night. These findings suggest that individuals did not experience substantially different levels of prey availability between years of the study.

The mean dive rate observed in the present study (1127 m h$^{-1}$) was substantially greater than in the epipelagic foraging subantarctic and Antarctic fur seals (346–372 m h$^{-1}$) [73,107] but similar to the benthic/demersal foraging conspecific Australian fur seal (*ca* 1000–1130 m h$^{-1}$, [49,87]). While no relationship was observed in the present study between the amount of benthic diving and dive rate, the relatively high dive rate suggests individuals expended greater effort in search of prey than purely epipelagic foragers, and is consistent with benthic/demersal foraging being more demanding [108]. Indeed, in a study of female New Zealand fur seals where up to 40% of dives were considered benthic/ demersal, the dive rate was 610 m h$^{-1}$ [47]. While none of the measured morphometric variables were found to influence dive rate in the present study, the negative relationship observed between BCI and dive rate suggests that more successful animals required less diving to acquire the necessary prey

resources. Relationships between female condition and foraging effort have also been shown for Antarctic fur seals [109], northern fur seals (Callorhinus ursinus, [110]) and Australian fur seals [101].

The Hct, Hb and Mb levels reported in the present study are similar to the levels observed in epipelagic foraging adult female California sea lions [111] but lower than in benthic foraging Galapagos (Zalophus wollebaeki), New Zealand (Phocarctos hookeri) and Australian sea lions [68,112]. Likewise, the mass-specific plasma and blood volumes of Cape fur seals were lower than in benthic foraging otariid species but similar to the epipelagic foraging California sea lions Z. californianus [111,113]. Correspondingly, the total body oxygen store estimated in the present study (43.3 ml$O_2$ kg$^{-1}$) is typical of predominantly epipelagic foraging species and is consistent with a previously reported interspecific relationship between body oxygen stores and median dive duration [108].

Numerous previous studies of free-ranging otariid seals have calculated an aerobic dive limit (cADL) by dividing the estimated total body oxygen stores with an assumed diving metabolic rate [59,68,111,114]. Such computations are useful when the study animals display few or no anaerobic dives (i.e. sharply increased post-dive recovery durations, [57]). However, obtaining accurate estimates of diving metabolic rate for the Cape fur seal may not be possible and reliance on values from other species or age-classes can introduce biases. In the present study, in the absence of adequate estimates of diving metabolic rate in Cape fur seals, the aerobic dive limit was determined in 18 individuals where there was clear evidence of sharply increased post-dive recovery durations (bADL).

The mean bADL of Cape fur seals observed in the present study (3.85 min) is considerably longer (by a factor of 1.4–2.3) than the cADL calculated for the larger (by a factor of 1.4–2.0) Australian sea lion, New Zealand sea lion, California sea lion and conspecific Australian fur seal [108]. This observation suggests that, in the absence of a significantly more energetically efficient mode of diving in Cape fur seals, the previously reported cADL of other otariids may be underestimates, highlighting the need for accurate estimates of diving metabolic rate [114]. Dividing the mean estimate of total body oxygen stores determined in the present study with the mean observed bADL, a diving metabolic rate estimate of 11.2 ml$O_2$ kg$^{-1}$ min$^{-1}$ was obtained for adult female Cape fur seals. This value is less than the 16.23 ml$O_2$ kg$^{-1}$ min$^{-1}$ estimated from heart rate in free-ranging Antarctic fur seals [115] but within the range (11–16 ml$O_2$ kg$^{-1}$ min$^{-1}$) recorded by respirometry from captive animals performing single dives in Steller sea lions (Eumetopias jubatus) [116,117] and surface swimming in California sea lions (12 ml$O_2$ kg$^{-1}$ min$^{-1}$) [118].

Most dives that exceeded the bADL were benthic, although this proportion varied among individuals. The overall proportion of benthic dives had a strong influence on the proportion of dives that exceeded the bADL in those females for which bADL could be estimated. This is consistent with previous findings across pinnipeds that benthic foraging is associated with an increased occurrence of dives exceeding calculated aerobic dive limits and a greater foraging effort [114,115]. Any increases in the amount of benthic diving in Cape fur seals, therefore, might be expected to increase the amount of time individuals have to spend at the surface during foraging trips to recover from anaerobic metabolism [57] and, hence, potentially reduce their foraging efficiency. However, further studies are required to investigate the degree to which benthic diving in Cape fur seals is associated with changing conditions in the Benguela, and the nutritional content of prey consumed in such dives, to assess this issue.

# 5. Conclusion

The results of the present study have revealed that Cape fur seals in the Benguela display a predominantly epipelagic foraging mode, consistent with previous reports indicating a diet largely comprising small schooling fish. The previously unreported 'pursuit' dive profile suggests individuals may opportunistically target larger predatory fish species they encounter in the process, although further investigation is necessary to evaluate this hypothesis. It is unclear whether the unexpected degree of benthic foraging observed is representative of the species in the region or is in response to environmental conditions specific to the time of the study. To evaluate this, studies linking dive behaviour with habitat utilization would be helpful. The greater putative costs of benthic foraging, along with the predominantly epipelagic diving, highlight the reliance of Cape fur seals on small school prey and the potential detrimental effects on its population of environmental variation in the Benguela due to global change.

Ethics. All fieldwork was permitted by the Animal Ethics Committee of the then Department of Environmental Affairs and Tourism's Marine and Coastal Management branch, which at the time was the management authority of South Africa's marine and coastal environment (Ref: DEAT2006-06-23).

Data accessibility. Dive data are available from the Dryad Digital Repository: https://doi.org/10.5061/dryad. 7m5s110 [119].

Authors' contributions. S.P.K.: fieldwork, data processing and analyses, interpretation, lead writer drafting and revising article. D.P.C.: conception and design, fieldwork and critical revision. A.-L.H.: fieldwork and critical revision. P.G.H.K.: fieldwork, logistics and revision. W.H.O.: sourcing of funding, conception and design, fieldwork and revision. M.W.: laboratory processing and analyses. J.A.B.: modelling analyses, interpretation and critical revision. J.P.Y.A.: sourcing of funding, conception and design, fieldwork, analyses and interpretation, co-drafting and critical revision.

Competing interests. The author and co-authors of this manuscript have no competing interests.

Funding. The research was supported by an Australian Research Council grant (grant no. DP0664167—J.P.Y.A., D.P.C., M.W.), the previous Department of Environmental Affairs and Tourism of South Africa (S.P.K., P.G.H.K., W.H.O.) and South Africa's National Research Foundation and the Nelson Mandela University Postgraduate Research Scholarship (J.A.B.).

Acknowledgements. We thank Mike Meÿer, Steven McCue, Darrell Anders, Mdu Seakamela, Adrian Tordiffe, Helene de Nys, Corrina Pieterse-Downes and Silvia Mecenero for their assistance in the field. The assistance of Sebastian Luque with the processing and analysing of dive data is gratefully acknowledged.

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
