## [Reviewer comments · Royal Society Open Science]

Review History

RSOS-191369.R0 (Original submission)

Review form: Reviewer 1

Is the manuscript scientifically sound in its present form?

Yes

Are the interpretations and conclusions justified by the results?

Yes

Is the language acceptable?

Yes

Do you have any ethical concerns with this paper?

No

Have you any concerns about statistical analyses in this paper?

No

Recommendation?

Accept as is

Comments to the Author(s)

This manuscript provides the first detailed record of diving behavior in Cape fur seals off South Africa. The study uses a relatively large sample (32 animals and over 32,000 dives) to provide an excellent new record of the diving behavior of this important predator species.

The majority of animals seem to primarily rely on epipelagic foraging, as is typical in many other fur seals. Although, it is interesting to find that some individuals performed more benthic foraging, especially given the importance of this diving style for the Australian sub-species.

It was also interesting to see the identification of "pursuit" dives. While I agree with this interpretation, it would be good to confirm this using animal-borne cameras as the authors suggest in a future study.

The analysis performed here is rigorous and appropriate for these data. The manuscript is clearly written and the figures provide useful visuals.

Congratulations on an excellent study.

Review form: Reviewer 2**Is the manuscript scientifically sound in its present form?**

Yes

Are the interpretations and conclusions justified by the results?

Yes

Is the language acceptable?

Yes

Do you have any ethical concerns with this paper?

No

Have you any concerns about statistical analyses in this paper?

No

Recommendation?

Accept with minor revision (please list in comments)

Comments to the Author(s)

The manuscript provides the first detailed assessment of the at-sea foraging and diving behaviour of the Cape (South African) fur seal. The fact that this is a populous, ecologically and economically important top predator species in this ecosystem makes it both surprising and important that this is the first such comprehensive study undertaken. The importance of this benchmark work is evident to improve understanding of the ecology of the species and the regional ecosystem. Comparisons with other otariid species indicate some unique foraging

behaviour associated with this species in question. This adds to our collective knowledge related to the taxa.

The manuscript is generally very well written, although some instances of ambiguous language or loose grammar are present. I point some of these out in the specific comments below. The study is thorough and well referenced with good contextualisation. In general, I have very few comments and commend the authors on a well formulated study and communication thereof. Specific comments (note page and line numbers have been allocated a bit incomprehensibly – my apologies for confusion in my reference to specific areas of the ms):

- Pg 2 (summary) line 44: 'they' is ambiguous. The prey or predator or both? Rephrase.
- Summary last sentence is clumsy. Rephrase and clarify.
- Intro Line 23: should be 'ecologically'
- Intro line 24-28; Restructure as: "While many marine top predators including several sharks, seabirds and some marine mammal species have been declining at rapid rates worldwide [8,9], other species have been in recovery following population depletion due to past over-harvesting (10-13)."
- Line 39: delete 'While'. "It has a generalist...." And then full stop after seabirds. Starting next sentence with "Yet, two thirds....." There are several places in the manuscript where sentence structure can be slightly amended to improve readability. I do not make mention of all such instances.
- Line 48-49: "which removes ca 2 million tons of marine organisms" – seals? Or fisheries? Or both? Ambiguous.
- Page 4 of 41, line 16 starting "Correspondingly,...." Is a long clumsy sentence. Split and restructure.
- Materials and methods seems to be appropriate. Analytical techniques follow current practice.
- Results, tables and figures are well presented and appropriate.
- Discussion, Page 10 of 41 line 5: what about terrestrial threats? humans, terrestrial predators? Presumably, human culling activities historically took place during the day? So if they are leaving before sunrise, but primarily feeding at night, then what are they doing out at sea all day? More discussion required on this aspect.
- Same page line 27-37: language is clumsy here. Clarify this important point here.
- - end of the page A. gazelle should be gazella.
- Page 11 of 41, line 7: sentence is ambiguous as it stands it seems there are these new pursuit dives and another. Clarify.
- Line 55: other top predators dependent on sardines, or showing concomitant declines? Penguins? Some referencing / discussion needed here.

Decision letter (RSOS-191369.R0)

03-Sep-2019

Dear Dr Kirkman

On behalf of the Editors, I am pleased to inform you that your Manuscript RSOS-191369 entitled "Dive behaviour and foraging effort of female Cape fur seals *Arctocephalus pusillus pusillus*" has been accepted for publication in Royal Society Open Science subject to minor revision in accordance with the referee suggestions. Please find the referees' comments at the end of this email.

The reviewers and handling editors have recommended publication, but also suggest some minor revisions to your manuscript. Therefore, I invite you to respond to the comments and revise your manuscript.

- Ethics statement

- Data accessibility

<http://datadryad.org/submit?journalID=RSOS&manu=RSOS-191369>

- Competing interests

- Authors' contributions

- Acknowledgements

- Funding statement

Please ensure you have prepared your revision in accordance with the guidance at <https://royalsociety.org/journals/authors/author-guidelines/> -- please note that we cannot publish your manuscript without the end statements. We have included a screenshot example of

the end statements for reference. If you feel that a given heading is not relevant to your paper, please nevertheless include the heading and explicitly state that it is not relevant to your work.

Because the schedule for publication is very tight, it is a condition of publication that you submit the revised version of your manuscript before 12-Sep-2019. Please note that the revision deadline will expire at 00.00am on this date. If you do not think you will be able to meet this date please let me know immediately.

Please note that Royal Society Open Science charge article processing charges for all new submissions that are accepted for publication. Charges will also apply to papers transferred to Royal Society Open Science from other Royal Society Publishing journals, as well as papers

submitted as part of our collaboration with the Royal Society of Chemistry (<http://rsos.royalsocietypublishing.org/chemistry>).

on behalf of Dr Denise Greig (Associate Editor) and Kevin Padian (Subject Editor)
openscience@royalsociety.org

Associate Editor Comments to Author (Dr Denise Greig):

Thank you for the submission – the analyses are thorough and appropriate, the manuscript is well written, and I like all the figures. I have a few minor comments similar to those noted by reviewer #2.

Summary

Page 2, line 44. Like reviewer 2, I was not sure who “they” referred to. I was also confused by the word “function”. Please clarify what you mean here.

Page 2, line 54. “...benthic dives, which was prevalent...” should be “...benthic dives, which were prevalent...”

Intro

Page 4, line 34. I would add a comma and delete “in” and “s” to say “As adult females, otariid seals...”

Methods

Page 7, line 15. Missing a period between “...study” and “Blood Hb...”

Table 3. For the third Dive rate model, are the variables missing?

Figure 2. The caption says “The beanlines (solid) and the overall line (dotted) represent the median dive depth or duration per female and the overall median, respectively.” However, I did not see a dotted line indicating the overall median. Is it missing? I would also rephrase as “The beanlines (solid) represent the median dive depth or duration per female and the overall line (dotted) represents the overall median” because it is easier for the reader.

Reviewer comments to Author:

Reviewer: 1

Comments to the Author(s)

This manuscript provides the first detailed record of diving behavior in Cape fur seals off South Africa. The study uses a relatively large sample (32 animals and over 32,000 dives) to provide an excellent new record of the diving behavior of this important predator species.

The majority of animals seem to primarily rely on epipelagic foraging, as is typical in many other fur seals. Although, it is interesting to find that some individuals performed more benthic foraging, especially given the importance of this diving style for the Australian sub-species.

It was also interesting to see the identification of “pursuit” dives. While I agree with this interpretation, it would be good to confirm this using animal-borne cameras as the authors suggest in a future study.

The analysis performed here is rigorous and appropriate for these data. The manuscript is clearly written and the figures provide useful visuals.

Congratulations on an excellent study.

Reviewer: 2

Comments to the Author(s)

The manuscript provides the first detailed assessment of the at-sea foraging and diving behaviour of the Cape (South African) fur seal. The fact that this is a populous, ecologically and economically important top predator species in this ecosystem makes it both surprising and important that this is the first such comprehensive study undertaken. The importance of this benchmark work is evident to improve understanding of the ecology of the species and the regional ecosystem. Comparisons with other otariid species indicate some unique foraging behaviour associated with this species in question. This adds to our collective knowledge related to the taxa.

The manuscript is generally very well written, although some instances of ambiguous language or loose grammar are present. I point some of these out in the specific comments below. The study is thorough and well referenced with good contextualisation. In general, I have very few comments and commend the authors on a well formulated study and communication thereof. Specific comments (note page and line numbers have been allocated a bit incomprehensibly – my apologies for confusion in my reference to specific areas of the ms):

- Pg 2 (summary) line 44: ‘they’ is ambiguous. The prey or predator or both? Rephrase.
- Summary last sentence is clumsy. Rephrase and clarify.
- Intro Line 23: should be ‘ecologically’
- Intro line 24-28; Restructure as: “While many marine top predators including several sharks, seabirds and some marine mammal species have been declining at rapid rates worldwide [8,9], other species have been in recovery following population depletion due to past over-harvesting (10-13).”
- Line 39: delete ‘While’. “It has a generalist....” And then full stop after seabirds. Starting next sentence with “Yet, two thirds.....” There are several places in the manuscript where sentence structure can be slightly amended to improve readability. I do not make mention of all such instances.
- Line 48-49: “which removes ca 2 million tons of marine organisms” – seals? Or fisheries? Or both? Ambiguous.
- Page 4 of 41, line 16 starting “Correspondingly,....” Is a long clumsy sentence. Split and restructure.
- Materials and methods seems to be appropriate. Analytical techniques follow current practice.
- Results, tables and figures are well presented and appropriate.
- Discussion, Page 10 of 41 line 5: what about terrestrial threats? humans, terrestrial predators? Presumably, human culling activities historically took place during the day? So if they are leaving before sunrise, but primarily feeding at night, then what are they doing out at sea all day? More discussion required on this aspect.

- Same page line 27-37: language is clumsy here. Clarify this important point here.
- - end of the page A. gazelle should be gazella.
- Page 11 of 41, line 7: sentence is ambiguous as it stands it seems there are these new pursuit dives and another. Clarify.
- Line 55: other top predators dependent on sardines, or showing concomitant declines? Penguins? Some referencing / discussion needed here.

Author's Response to Decision Letter for (RSOS-191369.R0)

See Appendix A.

Decision letter (RSOS-191369.R1)

23-Sep-2019

Dear Dr Kirkman,

I am pleased to inform you that your manuscript entitled "Dive behaviour and foraging effort of female Cape fur seals *Arctocephalus pusillus pusillus*" is now accepted for publication in Royal Society Open Science.

on behalf of Dr Denise Greig (Associate Editor) and Kevin Padian (Subject Editor)
openscience@royalsociety.org

Appendix A

Dear Editor

Thank you for the opportunity to revise our manuscript. Our responses to the comments of Reviewer 2 and to Associate Editor Dr Denise Greig, are below. We have addressed all the comments except for a single comment by Reviewer 2, where we felt that expanding the discussion in line with his/her comment would not be appropriate, and we explain why. We hope that the responses will be found to be clear and acceptable.

all the best

Steve Kirkman (and on behalf of all co-authors)

Reviewer: 2

Comments to the Author(s)

The manuscript provides the first detailed assessment of the at-sea foraging and diving behaviour of the Cape (South African) fur seal. The fact that this is a populous, ecologically and economically important top predator species in this ecosystem makes it both surprising and important that this is the first such comprehensive study undertaken. The importance of this benchmark work is evident to improve understanding of the ecology of the species and the regional ecosystem. Comparisons with other otariid species indicate some unique foraging behaviour associated with this species in question. This adds to our collective knowledge related to the taxa.

The manuscript is generally very well written, although some instances of ambiguous language or loose grammar are present. I point some of these out in the specific comments below. The study is thorough and well referenced with good contextualisation. In general, I have very few comments and commend the authors on a well formulated study and communication thereof.

Specific comments (note page and line numbers have been allocated a bit incomprehensibly – my apologies for confusion in my reference to specific areas of the ms):

- Pg 2 (summary) line 44: 'they' is ambiguous. The prey or predator or both? Rephrase.

Response: We have replaced "...how they function" with "how the predators function in this role"

- Summary last sentence is clumsy. Rephrase and clarify.

Response: We have tried to shorten and make it clearer: "The greater putative costs of benthic diving highlights the potential detrimental effects to Cape fur seals of well-documented changes in the availability of epipelagic prey species in the Benguela". To do this we shifted the first mention of the Benguela region to earlier in the abstract (2nd sentence)

- Intro Line 23: should be 'ecologically'

Response: Have corrected, thanks.

- Intro line 24-28; Restructure as: "While many marine top predators including several sharks, seabirds and some marine mammal species have been declining at rapid rates worldwide [8,9], other species have been in recovery following population depletion due to past over-harvesting (10–13)."

Response: Have corrected as suggested.

- Line 39: delete 'While'. "It has a generalist...." And then full stop after seabirds. Starting next sentence with "Yet, two thirds....." There are several places in the manuscript where sentence structure can be slightly amended to improve readability. I do not make mention of all such instances.

Response: Have adjusted as suggested.

- Line 48-49: "which removes ca 2 million tons of marine organisms" – seals? Or fisheries? Or both? Ambiguous.

Response: Agreed. Now changed to:

"Consequently, there is ever-present concern regarding competition for resources between the seal population, which removes *ca* 2 million tons of marine organisms per year in South Africa and Namibia [17], and commercial fisheries [21,22]."

- Page 4 of 41, line 16 starting "Correspondingly,...." Is a long clumsy sentence. Split and restructure.

Response: We have now split the sentence in two:

"Correspondingly, there have been associated effects on the survival, abundance, distribution, feeding behaviour and diet of several top predator populations such as the African penguin (*Spheniscus demersus*), Cape gannet (*Morus capensis*) and Cape cormorant (*Phalacrocorax capensis*) in the southern Benguela [36–40]. Changes that have been documented include longer foraging trips, feeding on suboptimal food, reduced adult and chick survival, declines in abundance and eastward shifts in distribution."

- Materials and methods seems to be appropriate. Analytical techniques follow current practice.
- Results, tables and figures are well presented and appropriate.
- Discussion, Page 10 of 41 line 5: what about terrestrial threats? humans, terrestrial predators? Presumably, human culling activities historically took place during the day? So if they are leaving before sunrise, but primarily feeding at night, then what are they doing out at sea all day? More discussion required on this aspect.

Response: The reviewer raises an interesting question. While harvesting by humans did occur at the site, this ceased >30 years ago and, thus, is unlikely to have a current impact on the present population (i.e. several generations post-hunting). Other known terrestrial predators (jackals, hyenas and historically lions) do prey during the day but more so at night. Furthermore, their abundance in the area is low and they act more as scavengers than hunters. Hence, we feel adding more discussion on this topic would be highly speculative and not appropriate.

Please note that in the paragraph referred to here, I have made two minor corrections.

- Same page line 27-37: language is clumsy here. Clarify this important point here.

Response: We have reworded and think that it is now much clearer, thanks you.

"This they found contradictory to diet data from offshore feeding seals which showed that deep water hake (*Merluccius paradoxus*) was found in seal stomachs. It was expected that this species could only be caught at night by seals during their diel vertical migration. However, the lack of night diving in the dive data they reported led the authors to consider that deep water hake in the seal

stomachs must have come from animals scavenging from trawl catches of this commercially important species. By comparison, the present study (based on 32 animals) demonstrated a prevalence of diving at night, illustrating the risk of basing such suppositions upon limited sample sizes and supports the notion that Cape fur seals could indeed be hunting deep water hake.”

- - end of the page A. gazelle should be gazella.

Response: Corrected

- Page 11 of 41, line 7: sentence is ambiguous as it stands it seems there are these new pursuit dives and another. Clarify.

Response: We hope that this change addresses it:

“The occurrence of “pursuit” dives, the second most common dive type in the present study and a profile not previously reported in seals, could also indicate the capture of single larger prey.”

Response: Please note that further down, I have now divided this rather long paragraph into two, with the break occurring after [93].

- Line 55: other top predators dependent on sardines, or showing concomitant declines? Penguins? Some referencing / discussion needed here.

Response: We think the concern is addressed below:

“This is consistent with well documented shifts in the distributions of spawning stocks of sardine and anchovy in South Africa towards the east Agulhas Bank since the 1990s [32,33]. These changes have resulted in decreased availability of these prey species in the Southern Benguela for not only seals, but also seabirds such as the African penguin, Cape gannet and Cape cormorant, that are dependent on these prey [36–40].”

Associate Editor Comments to Author (Dr Denise Greig):

Thank you for the submission – the analyses are thorough and appropriate, the manuscript is well written, and I like all the figures. I have a few minor comments similar to those noted by reviewer #2.

Summary

Page 2, line 44. Like reviewer 2, I was not sure who “they” referred to. I was also confused by the word “function”. Please clarify what you mean here.

Response: We hope that this amendment works:

“While marine top predators can play a critical role in ecosystem structure and dynamics through their effects on prey populations, how the predators function in this role is often not well understood”

Page 2, line 54. “...benthic dives, which was prevalent...” should be “...benthic dives, which were prevalent...”

Response: We have modified it as below:

“However, most females also performed benthic dives, and benthic diving was prevalent in some individuals.”

Intro

Page 4, line 34. I would add a comma and delete “in” and “s” to say “As adult females, otariid seals...”

Response: Done, thanks

Methods

Page 7, line 15. Missing a period between “...study” and “Blood Hb...”

Response: Added thanks...

Table 3. For the third Dive rate model, are the variables missing?

Response: Variables are not missing. However, for that permutation of the model the dredge function selected no predictor effects. It does happen sometimes when there are no specific predictor effects (or combinations of these) that clearly explain the response variable. This is indeed the case for Dive rate as can be seen in the model averaged coefficients (Table S2, supplementary material in the submission).

Motivated in part by your question, we have opted to swap around Table S2 and the relevant table in the Results. Further rationale for this is that while the model selection table (Table 3 in the submission) gives us all the possible models (combinations of predictors), the model averaged coefficients table (Table S2 in the submission) gives us the relative importance of each predictor effect, and therefore provides more of a “final product”.

In doing this we realised that the order of Tables 2 and 3 were incorrect in the submission. So we have corrected this.

So the changes are (a) Table 2 becomes Table 3 and vice versa. (b) Table S2 of the submission becomes Table 2, and Table 2 (which was incorrectly called Table 3 in the submission) becomes Table S2. The necessary changes are made in the references to tables in the results (4th and 6th paragraphs), as well as for the Table captions - all using tracked changes. The tables themselves have been rearranged accordingly - without using tracked changes. Sorry for this bit of confusion, I hope it is clear now.

Figure 2. The caption says “The beanlines (solid) and the overall line (dotted) represent the median dive depth or duration per female and the overall median, respectively.” However, I did not see a dotted line indicating the overall median. Is it missing? I would also rephrase as “The beanlines (solid) represent the median dive depth or duration per female and the overall line (dotted) represents the overall median” because it is easier for the reader.

Response: Excellent, done!

Please note, the name of the first author’s affiliation has changed since the first submission, and his second affiliation was slightly misrepresented. Thus slight changes have been made.